# Catalysts Derived from Nickel-Containing Layered Double Hydroxides for Aqueous-Phase Furfural Hydrogenation

**Olga B. Belskaya** [1,*] , **Roman M. Mironenko** [1] , **Tatiana I. Gulyaeva** [1] , **Mikhail V. Trenikhin** [1] , **Ivan V. Muromtsev** [1] , **Svetlana V. Trubina** [2] , **Valentina V. Zvereva** [2] and **Vladimir A. Likholobov** [3]

[1] Center of New Chemical Technologies, Boreskov Institute of Catalysis, 54 Neftezavodskaya Street, 644040 Omsk, Russia; ch-mrm@mail.ru (R.M.M.); tangul-8790@ihcp.ru (T.I.G.); tremv@yandex.ru (M.V.T.); muromtseviv@gmail.com (I.V.M.)

[2] Nikolaev Institute of Inorganic Chemistry, 3 Acad. Lavrentieva Ave., 630090 Novosibirsk, Russia; svt@niic.nsc.ru (S.V.T.); zvereva@niic.nsc.ru (V.V.Z.)

[3] Boreskov Institute of Catalysis, 5 Acad. Lavrentieva Ave., 630090 Novosibirsk, Russia; likholobov47@mail.ru

[*] Correspondence: obelska@ihcp.ru; Tel.: +7-(3812)-670-474

**Abstract:** Changes in the structural and textural properties of NiAl-layered double hydroxides (LDHs) (with 2–4 molar ratios of metals) and state of nickel that occur in different steps of the synthesis of nickel catalysts were studied using XRD, thermal analysis, TPR, low-temperature nitrogen adsorption, XANES, EXAFS, and electron microscopy methods. Relations between nickel content, catalyst reduction conditions, state of nickel, and its catalytic properties were revealed. It was shown that the use of NiAl LDH as the catalyst precursor even at a high content of active metal allows for the obtaining of the dispersed particles of supported nickel that are active in the aqueous-phase hydrogenation of furfural. The catalyst activity and conversion of furfural were found to increase with elevation of the catalyst reduction temperature and the corresponding growth of the fraction of reduced nickel. However, a lower reduction temperature (500 °C) makes it possible to form smaller nickel particles with the size of 4–6 nm, and a high Ni content (Ni:Al = 4) can be used to obtain the active Ni@NiAlO$_x$ catalyst. Under mild reaction conditions (90 °C, 2.0 MPa), the furfural conversion reached 93%, and furfuryl alcohol was formed with the selectivity of 70%. Under more severe reaction conditions (150 °C, 3.0 MPa), complete conversion of furfural was achieved, and cyclopentanol and tetrahydrofurfuryl alcohol were the main hydrogenation products.

**Keywords:** nickel catalysts; layered double hydroxides; furfural hydrogenation

## 1. Introduction

Catalytic hydrogenation reactions play a key role in both the low-tonnage organic synthesis of fragrances, pesticides, pharmaceuticals, and dyes, and the large-tonnage production of plastic semi-products and synthetic fibers [1–4]. Palladium on carbon supports serves as the main catalytic system employed in around 75% of hydrogenation reactions. However, due to the high cost of palladium, the development of methods for the synthesis of efficient and cheaper catalysts not containing platinum group metals remains a topical problem. This refers primarily to the Ni-containing systems, which are applied in liquid-phase hydrogenation and high-temperature reactions, for example, in steam reforming of methane and methanol [5–8]. The main problem in the synthesis of supported nickel catalysts is to obtain highly dispersed particles of the active metal with a uniform size, which provide high selectivity of transformations. Thus, it was shown [9–12] that in the synthesis of Ni/Al$_2$O$_3$ catalyst by conventional impregnation, even at a low Ni content of around 0.5–1.0 wt %, it is impossible to prevent the formation of large supported nickel particles, which are low active. Under high-temperature conditions, such particles cannot provide the desired selectivity, and in the presence of hydrocarbon gas, they are able to form carbon fibers, which lead to deactivation of the catalyst and clogging of the reactor

tubes. It was found that the selection of a precursor and thermal treatment conditions is important for the formation of nickel particles with the desired size [13]. In some approaches, dispersed nickel particles can be obtained by the surfactant templating method using cationic, anionic, and non-ionic surfactants as the structure-regulating agents [5]; moreover, the choice of Ni-MOF as precursors can provide a high surface area and the formation of isolated small-sized Ni nanoparticles [14].

One of the most promising approaches to this problem is the use of nickel-containing layered double hydroxides (LDHs) as the catalyst precursor, which allows for the obtaining of the dispersed metal particles even at high nickel content. LDHs are a family of natural and synthetic materials with the general formula $[M(II)_{1-x}M(III)_x(OH)_2]\,[A_{x/n}^{n-}]\cdot mH_2O$, where M(II) and M(III) are divalent and trivalent cations, respectively; $A^{n-}$ is the anion; and $x$ commonly ranges from 0.2 to 0.33 [15]. The LDH structure consists of the brucite-like layers formed by $OH^-$ groups, which have the closest packing arrangement. Oxygen atoms in the hydroxide ions form a system of octahedral voids that are statistically filled with di- and trivalent metal cations. The presence of triply charged cations creates an excess positive charge of the layers, which is compensated by anions residing in the interlayer spaces, where water molecules are also present [15,16]. LDHs allow for a uniform distribution of two or more cations in the general crystal structure [16], which makes them excellent precursors for obtaining nonstoichiometric mixed oxides with the specified composition and crystallinity [17,18]. Thus, the NiAl mixed oxides obtained by calcination of carbonate-containing NiAl LDHs (takovite) demonstrate good catalytic properties, owing to the high dispersion of the particles and thermal stability. Such oxides retain their structure and properties up to 850 °C, i.e., the temperature at which stoichiometric spinel $NiAl_2O_4$ starts to form [19,20].

The synthesis of the catalyst based on nickel-aluminum LDHs, which commonly includes calcination and reduction steps, can formally be presented (e.g., for the sample with Ni:Al = 3) as

$$Ni_6Al_2(OH)_{16}CO_3\cdot4H_2O \rightarrow 6NiO + Al_2O_3 + 12H_2O + CO_2, \qquad (1)$$

$$NiO + H_2 \rightarrow Ni + H_2O. \qquad (2)$$

This scheme implies that the nickel present in NiAl LDH completely passes into NiO upon calcination, from which metallic nickel is formed in the reduction step. However, it was found that thermal decomposition of takovite leads to the formation of two types of mixed oxides consisting of alumina, which contains poorly crystallized bunsenite (NiO), and nickel oxide with alumina impurity [19]. Our earlier study [18], where the differential dissolution method was used to separate phases and reveal their composition, showed the presence of oxide phases with the metal ratio $Al_1Ni_{0.3}$ and $Al_{0.24}Ni_1$, the fraction of which was equal, respectively, to 16.4 and 83.6% (for Ni:Al = 2). Modeling of X-ray diffraction (XRD) patterns for one-dimensionally disordered crystals demonstrated that the oxide phase enriched with nickel is the layered defect spinel, while the aluminum-enriched phase is X-ray amorphous; however, the presence of exactly the latter phase determines the behavior of nickel-containing oxides, particularly their difficult restructuring upon contact with water.

After the reductive treatment, Ni(II) cations give the metallic Ni phase dispersed in the oxide matrix [21–23]. Such catalysts were reported to be efficient, for example, in the gas-phase hydrogenation of acetylene compounds [24], as well as catalytic upgrading of biomass-derived carbonyl compounds [25–27]. It should be noted that, although hydrogenation catalysts obtained by sequential thermal and reductive treatments of co-precipitated NiAl hydroxides have been known for about 100 years (the so-called Zelinsky catalyst [28]), interest in their study and improvement is still unabated [25–27,29–32]. The present work develops the approach indicated above to investigate the transformation of Ni-containing LDHs in the catalyst synthesis. The goal was to establish the formation regularities of nickel sites in dependence on the ratio between doubly and triply charged

cations in nickel–aluminum LDHs, as well as the conditions of reductive treatment. The synthesized catalysts were studied in the aqueous-phase hydrogenation of furfural (FAL) (FAL is considered as a representative of platform molecules). The use of mild aqueous-phase reaction conditions is important for processing of vegetable feedstock and initiates the development of advanced, efficient, and more available and non-noble metal-based catalysts [33].

## 2. Results and Discussion

### 2.1. Synthesis of NiAl LDHs and Mixed Oxides

Among various methods of LDH synthesis, precipitation from solutions has some advantages: simplicity and efficiency of the process and high purity of the product [15,16]. Co-precipitation of nickel and aluminum hydroxides upon hydrolysis of their salts was used to obtain NiAl LDHs, which are the synthetic analogs of the natural material takovite. Takovite belongs to a family of anionic clays and contains mixed positively charged nickel–aluminum hydroxide layers separated by interlayer spaces that are formed from carbonate anions and water molecules. The chemical composition and main structural characteristics of the synthesized $NiAl-CO_3$ samples, which have the Ni/Al atomic ratio in the range of 2–4, are listed in Table 1. In all the samples, the measured ratio of metals correspond to the required value, which testifies to a high completeness of the hydrolysis.

**Table 1.** Composition and structural characteristics of $NiAl-CO_3$.

| Calculated Ni/Al Atomic Ratio | Content, wt % [1] | | | Measured Ni/Al Atomic Ratio [1] | $c$, Å [2] | $a$, Å [2] | $L_c$, nm [2] | $L_a$, nm [2] |
|---|---|---|---|---|---|---|---|---|
| | Ni | Al | O | | | | | |
| 2 | 53.3 | 12.2 | 34.5 | 2.1 | 22.745 | 3.021 | 48.8 | 77.9 |
| 3 | 62.2 | 8.5 | 29.3 | 3.0 | 23.232 | 3.043 | 71.1 | 116.7 |
| 4 | 64.2 | 6.9 | 28.9 | 4.1 | 23.478 | 3.054 | 48.6 | 102.6 |

[1] According to inductively coupled plasma atomic emission spectroscopy (ICP-AES) and atomic absorption spectrometry (AAS). [2] According to X-ray diffraction (XRD).

According to XRD data (Figure 1a), the structure of all the samples is characterized by the presence of a series of basal reflections 003 and 006 as well as the peaks of the {0*kl*} family: 012, 015, 018; {*hk*0}: 110, and the 113 peak, and corresponds to the hydrotalcite structure (no. 22-700, ICDD, PDF-2). This structure is preserved over the entire range of Ni/Al atomic ratios. Signals from other crystal phases were not detected. The ratio of doubly and triply charged cations in hydroxide layers is the main factor determining the capacity of interlayer space and the strength of interaction between hydroxide layers and interlayer anions. According to the data of Table 1, lattice parameters $c$ and $a$ monotonically increase with increasing the molar fraction of $Ni^{2+}$. The growth of parameter $c$ is related to the weakening of electrostatic interaction between positively charged brucite-like layers and interlayer spaces, which is caused by a decrease in the positive charge (a decrease in the $Al^{3+}$ fraction) of the brucite-like layer. The growth of the lattice parameter $a$, which characterizes the shortest distance between two cations in the layer, results from an increase in the fraction of $Ni^{2+}$ cations, which have a greater ionic radius (0.068 nm) as compared with the ionic radius of aluminum (0.0535 nm). It should be noted that no clear dependence was observed between changes in the crystallite sizes and variation of the cationic ratio along $c$ ($L_c$) and $a$ ($L_a$) directions.

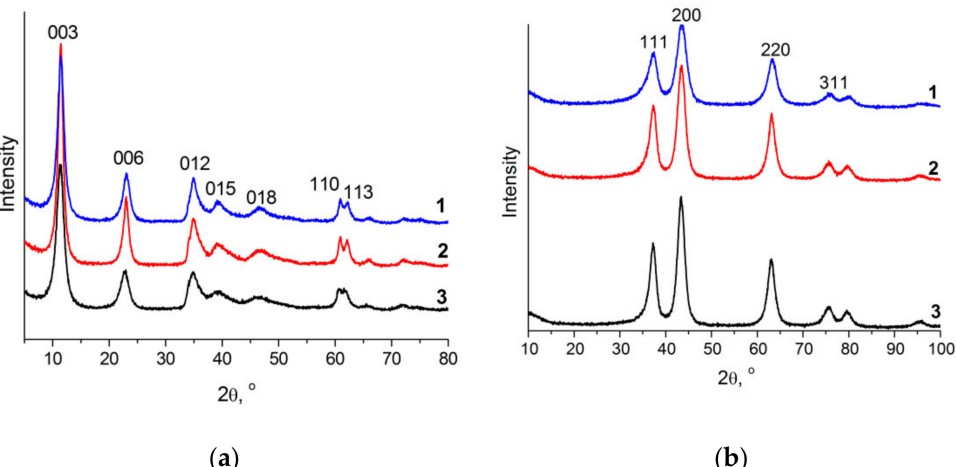

**Figure 1.** Diffraction patterns of NiAl LDH samples in the carbonate form (**a**) and the corresponding oxide phases (**b**) with the Ni/Al atomic ratio of 2 (line 1), 3 (line 2), and 4 (line 3).

NiAl LDHs possess a spongy morphology; in distinction to LDHs of different composition [34], their morphology remains virtually unchanged upon calcination of takovite to the corresponding mixed oxide (Figure 2). Thermal oxidative treatment of LDHs is used to obtain highly dispersed polymetallic oxide catalysts or catalyst supports with a high surface area; in the case of NiAl LDHs, such treatment is also a step in the synthesis of Ni@NiAlO$_x$ catalysts containing reduced nickel as the active component. Thermal analysis of NiAl LDH samples with different Ni/Al ratio in an air medium revealed two weight loss regions: up to 260 °C (the reversible removal of adsorbed and interlayer water) and the high-temperature region (irreversible dehydroxylation of the layers and decomposition of the interlayer carbonate ions) (Figure S1) [18,22]. As the fraction of the doubly charged cation increases, the maximum of the low-temperature peak broadens and shifts toward lower temperatures, which indicates a weakening of the interaction strength of hydroxide layers, whereas positions of the maxima of high-temperature peaks remain virtually unchanged (320–325 °C) [8,21,35]. So, for all the samples, the oxide phase was obtained at a similar temperature of oxidative treatment, 600 °C, at which the destruction of the layered structure terminates without the formation of stoichiometric spinel phases.

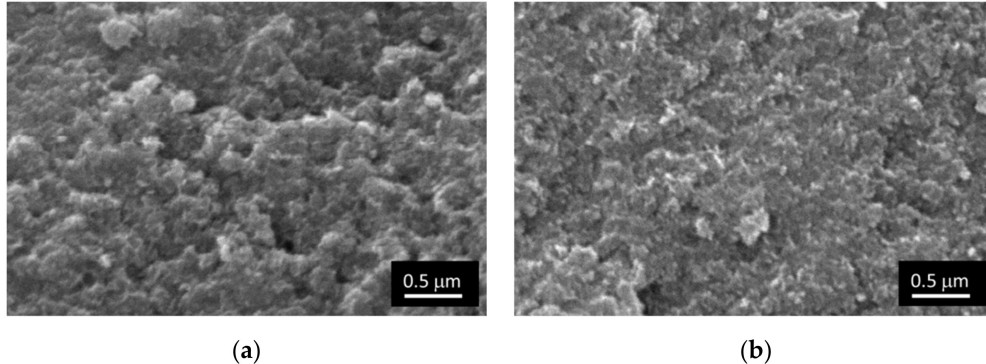

(**a**)

(**b**)

**Figure 2.** SEM images of NiAl-CO$_3$ with Ni/Al = 3 (**a**) and NiAlO$_x$ with Ni/Al = 3 (**b**).

Under the given calcination conditions, diffraction patterns of the samples contain only the peaks that have close positions to the peaks of the bunsenite-type phase (NiO) (Figure 1b) and characterize NiAlO$_x$ mixed oxide phase [18,21,22]. For all the samples, the obtained low-temperature nitrogen adsorption isotherms are typical of mesoporous objects (Figure S2). The produced oxides have a developed texture and a uniform pore size distribution in the region of $d$ = 3–20 nm with the average diameter 8–9 nm (Figure S3). The absence of micropores is an essential factor for the liquid-phase reactions because

it eliminates diffusion limitations. Specific surface area of the obtained mixed oxides was 180 $m^2 \cdot g^{-1}$ for the samples with Ni/Al = 2; 3 and 120 $m^2 \cdot g^{-1}$ for the sample with Ni/Al = 4 (Table S1).

### 2.2. Formation of Catalytically Active Phase upon Reductive Treatment

The formation of metallic nickel from nickel–aluminum oxides $NiAlO_x$ was studied using temperature-programmed reduction (TPR). The analysis of TPR profiles revealed the effect exerted by the Ni/Al atomic ratio on the position and shape of the peak (Figure 3). The profiles obtained in our study had a single distinct maximum, whereas Titulaer et al. [36] observed three temperature maxima of hydrogen consumption at 450, 550, and 750 °C upon reduction of takovite, which were attributed to the reduction of individual phases having different nickel content (the reduction of the dispersed NiO phase, the defect spinel phase formed from NiO and a certain amount of aluminum cations, and nickel in nickel-containing alumina, respectively). According to Figure 3, as the fraction of nickel increased, the fraction of hydrogen consumed in the low-temperature region also increased, which produced changes in the shape of the peak and position of the maximum. The hydrogen consumption maximum shifted from 600 °C for $NiAlO_x$-2 and $NiAlO_x$-3 (Ni:Al = 2 and 3) to 500 °C for $NiAlO_x$-4 (Ni:Al = 4). Thus, an increase in the fraction of nickel-enriched oxide phase weakened the Ni–Al interaction, facilitated the reduction of metal, and made it possible to obtain metallic nickel under milder conditions. Data on the amount of consumed hydrogen and the degree of nickel reduction depending on the composition of mixed oxide samples and TPR conditions are given in Table S2. For the most "high-temperature" sample with the minimum nickel content (Ni/Al = 2), measurements of the hydrogen amount consumed in the TPR experiment showed that upon reduction at 550 °C (holding for 1 h), the reduction degree of nickel did not exceed 55% (according to Equation (2)). When TPR was performed up to 900 °C, as in the case of holding at 650 °C, hydrogen consumption was close to stoichiometric for the complete reduction of metal.

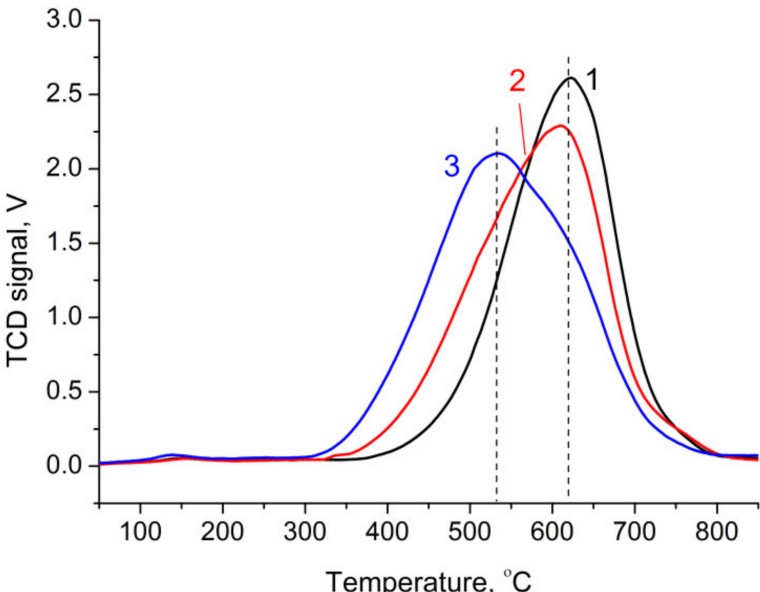

**Figure 3.** TPR profiles of $NiAlO_x$-2 (line 1), $NiAlO_x$-3 (line 2), and $NiAlO_x$-4 (line 3). Pre-calcination temperature was 600 °C.

Thus, the control of reduction temperature allows for the controlling of the amount of metallic nickel and, accordingly, the properties of the synthesized catalyst. XRD and EXAFS were used to quantitatively estimate the ratio of oxidized and reduced nickel species in dependence on the ratio of metals and reduction conditions. Figure 4 displays XRD patterns of the samples differing in both the Ni/Al atomic ratio and their reduction temperature.

One can see that the intensity of peaks of the crystal nickel phase increased monotonically with increasing the Ni content in the samples and elevating the reduction temperature. Data on the quantitative ratio of reduced nickel and nickel in the oxide phase as well as the crystallographic characteristics of these phases are listed in Table 2. Variation of the Ni/Al atomic ratio at a fixed reduction temperature of 500 °C led to a monotonic growth of the reduced nickel fraction from 27 to 39%. Therewith, dispersed nickel particles with the size of 4–6 nm were formed in all the samples. The lattice parameter of the nickel oxide phase also had a close value of 0.4135–0.4156 nm, which testifies to its nearly similar chemical composition. The lower values of lattice parameters in comparison with those for bunsenite (0.4177 nm) were related to the presence of aluminum cations with a smaller ionic radius in the oxide phase.

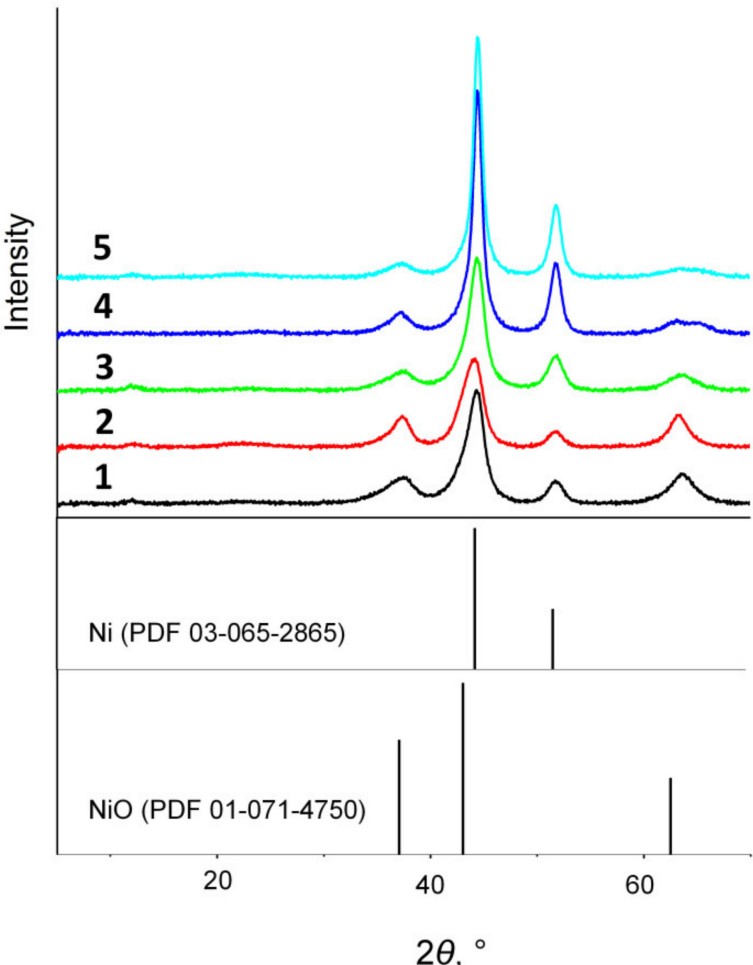

**Figure 4.** Diffraction patterns of the Ni@NiAlO$_x$ samples with different Ni/Al atomic ratio and obtained at different reduction temperatures (holding at the reduction temperature for 2 h): Ni@NiAlO$_x$-2-500 (line 1), Ni@NiAlO$_x$-3-500 (line 2), Ni@NiAlO$_x$-4-500 (line 3), Ni@NiAlO$_x$-3-600 (line 4), Ni@NiAlO$_x$-4-600 (line 5).

**Table 2.** Microstructural parameters of the Ni@NiAlO$_x$ samples with different atomic ratio of metals and obtained at different reduction temperatures (according to XRD measurements).

| Sample | Phase | Fraction, % | Unit Cell Parameters, nm | CSR, nm [1] |
|---|---|---|---|---|
| Ni@NiAlO$_x$-2-500 | Ni | 27 | 0.3532 | 5.2 |
| | NiO | 73 | 0.4135 | 2.4 |
| Ni@NiAlO$_x$-3-500 | Ni | 30 | 0.3538 | 3.6 |
| | NiO | 70 | 0.4156 | 3.7 |
| Ni@NiAlO$_x$-4-500 | Ni | 39 | 0.3533 | 5.7 |
| | NiO | 61 | 0.4136 | 2.5 |
| Ni@NiAlO$_x$-3-600 | Ni | 45 | 0.3529 | 10.9 |
| | NiO | 55 | 0.4122 | 3.4 |
| Ni@NiAlO$_x$-4-600 | Ni | 49 | 0.3528 | 9.7 |
| | NiO | 51 | 0.4116 | 3.0 |

[1] Coherent scattering region.

An increase in the reduction temperature of samples with the Ni/Al atomic ratio of 3 and 4 not only increased the fraction of metallic nickel from 30 and 39% to 45 and 49%, respectively, but also increased their particle size up to about 10 nm. A further transition of nickel into the metallic phase was also accompanied by a decrease in the lattice parameter of the remaining oxide phase to 0.412 nm due to increasing aluminum fraction (a decrease in the Ni/Al atomic ratio).

A similar trend was observed in the modeling of XANES and EXAFS spectra (Figures 5 and 6). For the Ni@NiAlO$_x$-2-500 and Ni@NiAlO$_x$-3-500 samples (Figure 5a), the intensity of white line, which characterized the density of unoccupied 3d states of nickel, increased due to a high fraction of the oxide phase. White line intensity (WLI) for the Ni@NiAlO$_x$-4-500 sample containing a greater amount of nickel, and also for the samples reduced at 600 °C (Figure 5b), was low, which indicates that the metallic phase was predominant in the samples. Figure 6, showing the Fourier-transformed EXAFS, reflects the same tendency concerning nickel oxide and metal fractions in the samples.

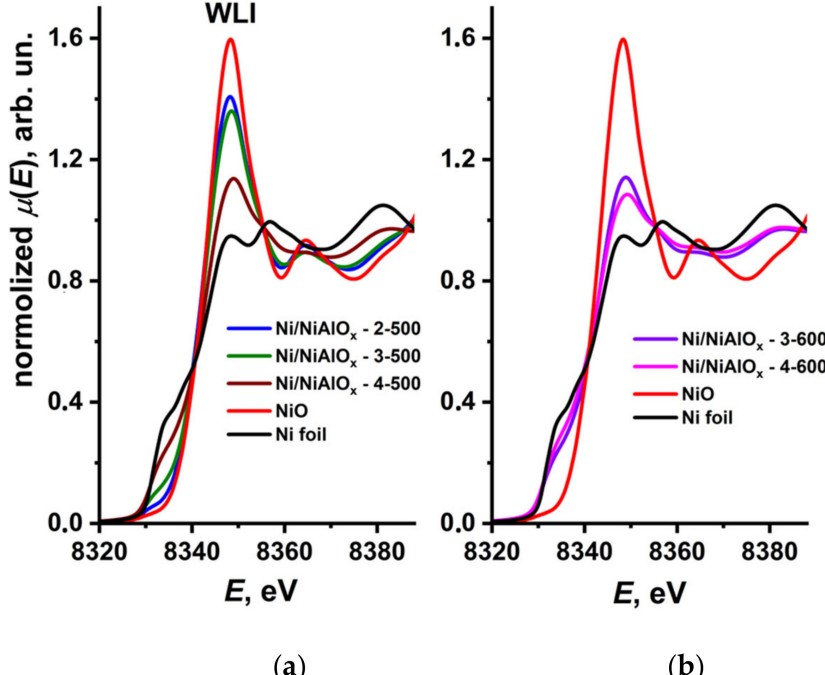

(**a**)          (**b**)

**Figure 5.** Ni K XANES spectra of the nickel–aluminum oxides reduced at 500 °C (**a**) and at 600 °C (**b**), as well as of the reference compounds (red line for NiO and black line for Ni foil).

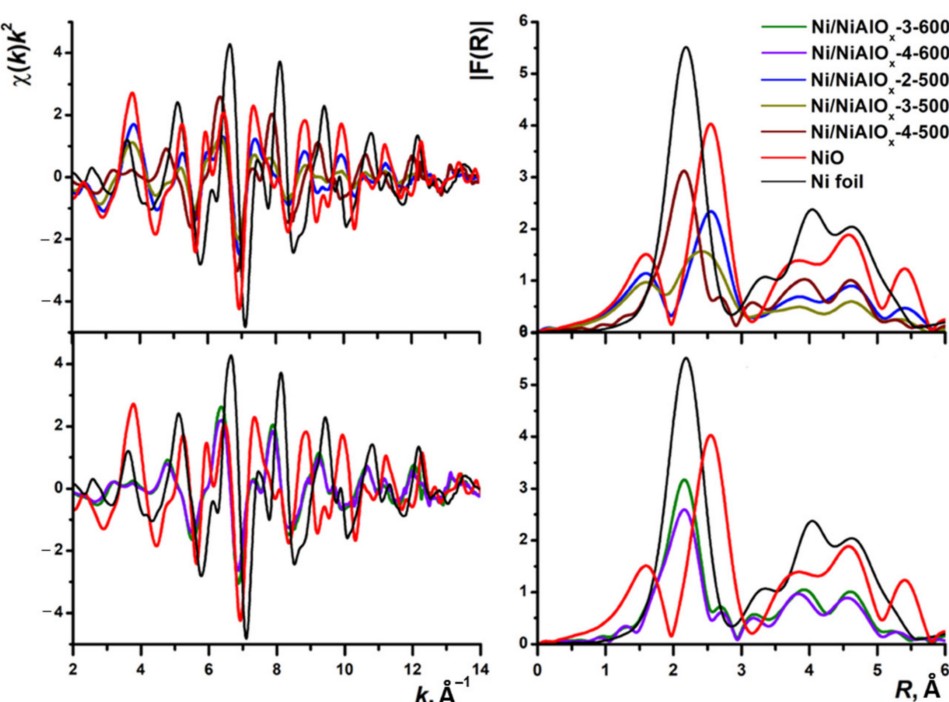

**Figure 6.** Ni K EXAFS spectra (**left**) of the reduced nickel–aluminum oxides and their Fourier transform moduli (without phase shift correction) (**right**). Spectra of the reference compounds: NiO (red line) and Ni foil (black line).

The modeling of EXAFS spectra took into account the two-phase nature of the samples with the separation into NiO and Ni metal components. Upon modeling, parameters of individual phases (coordination numbers and interatomic distances) were specified according to crystallographic data. As shown in Table 3, the fraction of reduced nickel increased with the growth of Ni/Al atomic ratio in the sample, and even to a greater extent with elevation of the reduction temperature; for the Ni@NiAlO$_x$-4-600 sample with the maximum content of nickel, the fraction of reduced nickel can achieve 70%.

Although the quantitative data on the phase ratio obtained by XRD and EXAFS may have differed due to limitations of each method (the presence of X-ray amorphous phases, the use of different models for treatment of diffraction patterns and EXAFS spectra), a general trend related to the effect of chemical composition and reduction conditions on the formation of supported particles of metallic nickel was retained.

The segregation of metallic nickel from the mixed oxide structure produced changes in the texture of the samples. A decrease in specific surface area was observed. For example, in a series of samples with Ni/Al = 2, upon reduction at 500 and 600 °C, the specific surface area decreased from 180 (before reduction) to 170 and 130 m$^2 \cdot$g$^{-1}$, respectively (Table S1). As follows from the analysis of the pore size distribution curves, such a decrease in the surface area is related to a decrease in the fraction of pores having a smaller diameter. With elevation of the reduction temperature, the distribution became narrower, and the maximum shifted toward larger pores (Figure 7). The average value of pore diameter increased to 13 nm for the samples reduced at 600 °C. Such a decrease in the amount of smallest mesopores at a slight decrease in the surface area of samples can be expected to have a positive effect in the case of liquid-phase reactions, because it may decrease diffusion limitations.

**Table 3.** Modeling of the Ni K EXAFS spectra with separation into two phases for the samples differing in the Ni/Al atomic ratio and reduction temperature. The accuracy upon decomposition into phases was ≈10%. Fit index shows the goodness of fit.

| Sample | NiO | | | | Ni Metal | | | |
|---|---|---|---|---|---|---|---|---|
| | Fraction, % | Coordination Sphere | $R$, Å [1] | $N$ [2] | Fraction, % | Coordination Sphere | $R$, Å [1] | $N$ [2] |
| Ni@NiAlO$_x$-2-500 | 90 | Ni–O<br>Ni–Ni | 2.09<br>2.97 | 6<br>12 | 10 | Ni–Ni | 2.48 | 12 |
| | | Fit = 2.6 | | | | | | |
| Ni@NiAlO$_x$-3-500 | 70 | Ni–O<br>Ni–Ni | 2.10<br>2.97 | 6<br>12 | 30 | Ni–Ni | 2.47 | 12 |
| | | Fit = 2.9 | | | | | | |
| Ni@NiAlO$_x$-4-500 | 38 | Ni–O<br>Ni–Ni | 2.13<br>2.99 | 6<br>12 | 62 | Ni–Ni | 2.48 | 12 |
| | | Fit = 1.6 | | | | | | |
| Ni@NiAlO$_x$-3-600 | 35 | Ni–O<br>Ni–Ni | 2.13<br>3.00 | 6<br>12 | 65 | Ni–Ni | 2.48 | 12 |
| | | Fit = 1.1 | | | | | | |
| Ni@NiAlO$_x$-4-600 | 30 | Ni–O<br>Ni–Ni | 2.13<br>3.00 | 6<br>12 | 70 | Ni–Ni | 2.49 | 12 |
| | | Fit = 2.3 | | | | | | |
| **Crystallographic data for NiO** | | | | | | | | |
| Coordination spheres | | | $R$, Å [1] | | | | $N$ [2] | |
| Ni–O | | | 2.08 | | | | 6 | |
| Ni–Ni | | | 2.94 | | | | 12 | |
| **Crystallographic data for Ni metal** | | | | | | | | |
| Coordination sphere | | | $R$, Å [1] | | | | $N$ [2] | |
| Ni–Ni | | | 2.49<br>3.52<br>4.32 | | | | 12<br>6<br>24 | |

[1] Interatomic distance (the measurement accuracy ±0.01 Å). [2] Coordination number (was fixed when modeling the separation into phases).

A transmission electron microscopy (TEM) study of the reduced samples (Figure 8) demonstrated that they had a similar morphology and consist of agglomerated particles with a predominantly spherical or arbitrary shape, the size of which varied in the range of 5–20 nm. The analysis of individual particles allowed us to identify nickel in the samples in the composition of both the oxidized and reduced species, which is consistent with XRD and EXAFS data. The crystal lattice of the dark particles observed in TEM images had the interplanar distance of 0.203 nm, which is typical of (011) facet of metallic nickel (PDF 45-1027), while brighter particles exhibited the interplanar distance of 0.208 nm, which belonged to the (012) facet of the NiO phase (PDF 44-1159). Calculations of interplanar distances in nanoparticles were carried out using the Digital Micrograph "Gatan" software after fast Fourier transform (FFT) processing of TEM images (Figure S4). Unfortunately, due to superposition of the particles, it is impossible to estimate the particle size of reduced nickel from electron microscopy data. However, a comparison of coherent scattering regions (Table 2) showed that elevation of the reduction temperature from 500 to 600 °C produced almost a twofold increase in the nickel crystallite size.

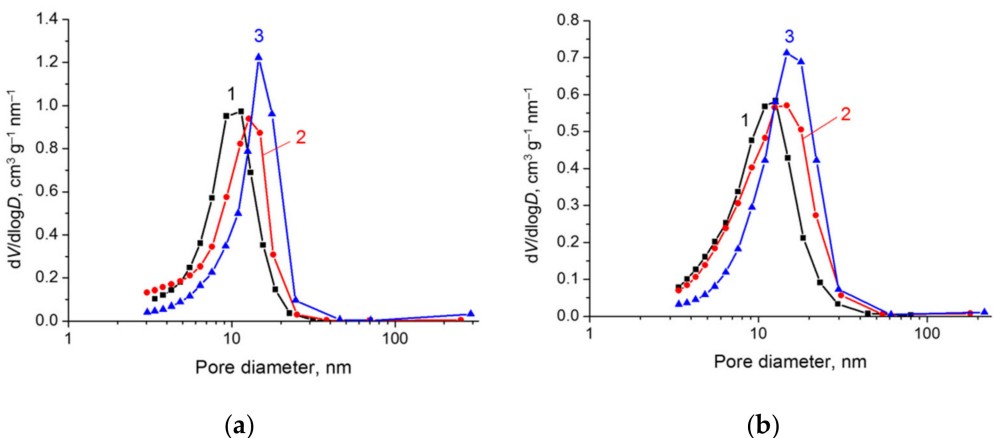

(**a**)                    (**b**)

**Figure 7.** The pore size distribution curves plotted from the $N_2$ adsorption branch for the samples with different Ni/Al atomic ratio: (**a**) Ni/Al = 2; (**b**) Ni/Al = 4. Line 1 corresponds to $NiAlO_x$ obtained by calcination of NiAl LDH at 600 °C, line 2 corresponds to Ni@$NiAlO_x$ obtained by calcination of NiAl LDH at 600 °C and subsequent reduction with $H_2$ at 500 °C, and line 3 corresponds to Ni@$NiAlO_x$ obtained by calcination of NiAl LDH at 600 °C and subsequent reduction with $H_2$ at 600 °C.

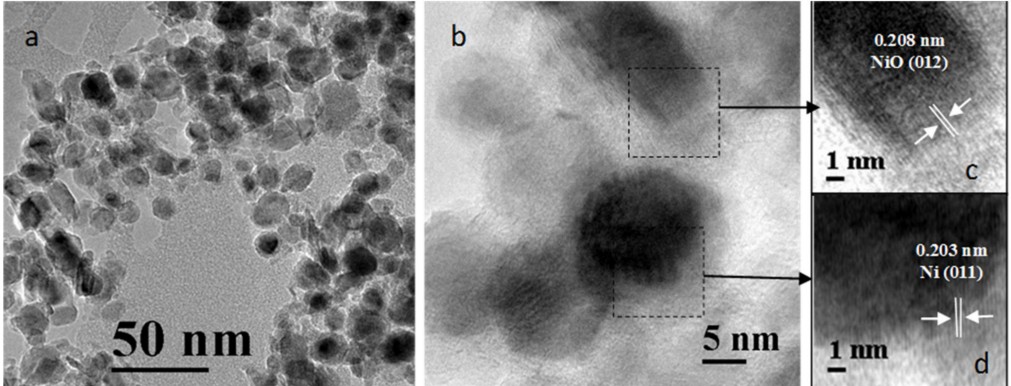

**Figure 8.** TEM images of the Ni@$NiAlO_x$-4-600 sample at different magnifications (**a**,**b**); crystal lattice of NiO (**c**) and Ni (**d**) nanoparticles.

The energy dispersive X-ray (EDX) analysis carried out for different regions of the Ni@$NiAlO_x$-4-600 sample demonstrated (Figure S5, Table S3) that nickel was distributed quite uniformly. The Ni/Al atomic ratios were in the range of 3.0–4.3, and the average value was 3.8, which was close to that determined by chemical analysis (Table 1). At the same time, in the region slightly enriched with aluminum (region 2), the Ni/O atomic ratio was equal to 0.4, and nickel was probably in the composition of mixed oxide. Regions of the images containing dark particles (for example, region 3) at the Ni/Al ratio close to 4, on the contrary, had a high value of Ni/O = 1.4, which testifies to a higher content of metallic nickel.

### 2.3. Performance of Ni@$NiAlO_x$ Catalysts in Aqueous-Phase Hydrogenation of Furfural (FAL)

The effect exerted by the composition of nickel catalysts synthesized from NiAl LDHs and their activation conditions on the catalytic properties was studied in the aqueous-phase hydrogenation of FAL. The study was carried out both under relatively mild reaction conditions (at 90 °C, 2.0 MPa), when mainly hydrogenation products such as furfuryl alcohol (FOL) and tetrahydrofurfuryl alcohol (THFOL) are produced [37–39], and under more severe conditions (150 °C, 3.0 MPa), at which water participates in catalytic conversions resulting in a wider range of products [37,40,41] (Scheme 1).

**Scheme 1.** Reaction network for the aqueous-phase hydrogenation of FAL over the Ni@NiAlO$_x$ catalysts. CPONE and CPOL were formed with a noticeable yield under harsh hydrothermal conditions (150 °C, 3.0 MPa) when water was involved in catalytic conversions; FOL and THFOL are the main products under milder reaction conditions (*cf*. Tables 4 and 5 below).

Under mild reaction conditions, a preliminary study of the nickel catalysts synthesized by conventional impregnation of γ-alumina and magnesium–aluminum oxide (10%Ni/γ-Al$_2$O$_3$ and 10%Ni/MgAlO$_x$) showed that these samples were inactive. Moreover, even a considerable increase in the nickel content (50%Ni/γ-Al$_2$O$_3$) led to the formation of the catalyst with a very low activity, which rapidly deactivated (Figure S6). As was noted earlier, this may have been caused by the formation of coarse nickel particles. By way of example, Figure S7 displays a TEM image of the 10%Ni/MgAlO$_x$ sample, which had a denser morphology and contained agglomerated particles of the supported metal. As follows from the data of Figure S8 and Table 4, under mild reaction conditions, the unreduced NiAlO$_x$ sample also did not exhibit the hydrogenating activity, and conversion values for Ni@NiAlO$_x$ samples were related to the fraction of metallic nickel in them. Samples with Ni/Al = 2 and 3, which were reduced at 500 °C, demonstrated very close catalytic properties (reaction rate, FAL conversion, and selectivity to FOL and THFOL). At about 40% conversion of FAL, the formation of FOL proceeded with a high selectivity (84%). For the Ni@NiAlO$_x$-4-600 sample with the maximum fraction of reduced nickel, the conversion of FAL was almost complete when the reaction was performed for 90 min.

**Table 4.** Catalytic measurements in hydrogenation of FAL for Ni@NiAlO$_x$ samples differing in the Ni/Al atomic ratio and reduction temperature.

| Catalyst [1] | Conversion of FAL, % [2] | Selectivity, % [3] | | Reaction Rate, (mmol H$_2$)·min$^{-1}$ [4] |
|---|---|---|---|---|
| | | FOL | THFOL | |
| Ni@NiAlO$_x$-2-500 | 39 | 84 | 13 | 22.7 |
| Ni@NiAlO$_x$-3-500 | 43 | 84 | 11 | 22.1 |
| Ni@NiAlO$_x$-4-500 | 93 | 70 | 25 | 38.5 |
| Ni@NiAlO$_x$-3-600 | 75 | 71 | 21 | 22.1 |
| Ni@NiAlO$_x$-4-600 | 98 | 52 | 27 | 33.0 |
| NiAlO$_x$-3 (unreduced) | 0 | – | – | – |

[1] Reaction conditions: 0.5 g catalyst, 5.0 cm$^3$ FAL, 100 cm$^3$ water, 90 °C, 2.0 MPa. [2] According to GC (after the completion of the reaction and the absence of hydrogen consumption). [3] According to GC. The reaction products also included small amounts of tetrahydrofurfural and the interaction products of FAL with the solvent (identified by NMR spectroscopy). [4] The reaction rate was calculated on the initial linear segment of kinetic curve as the amount of hydrogen consumed per minute during hydrogenation of FAL.

For all the samples, the main conversion product was FOL; with a growth of FAL conversion, selectivity to THFOL reached 27% for the most active catalyst. In general, as can be clearly seen from Figure 9a, with an increase in the fraction of metallic Ni in Ni@NiAlO$_x$ from 27 to 49% (according to XRD data), the FOL selectivity decreased from 84 to 52%, while the THFOL selectivity increased from 13 to 27%. It should be noted that the reaction rate, which was measured on the linear section of kinetic curve, had the maximum value for the Ni@NiAlO$_x$-4-500 sample, which, according to XRD data, at a considerable fraction of metallic nickel (around 40%) was characterized by the dispersed state of supported metal. Therewith, at a FAL conversion of 93%, this sample had a high (70%) selectivity to FOL.

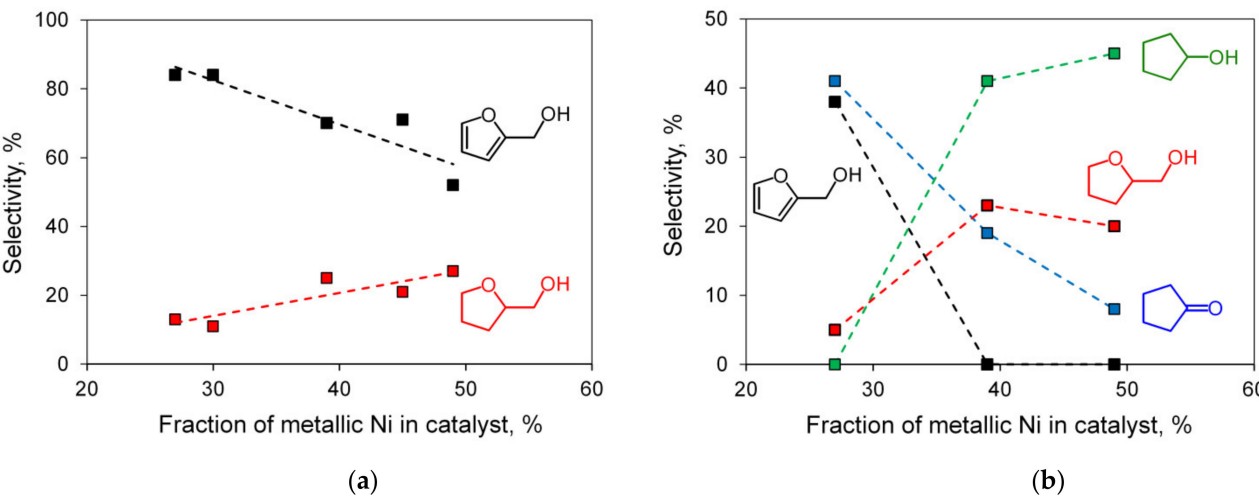

**Figure 9.** Selectivity to the hydrogenation products vs. fraction of metallic Ni in Ni@NiAlO$_x$ system (according to XRD data; see Table 2) for the FAL hydrogenation at 90 °C, 2.0 MPa (**a**), and at 150 °C, 3.0 MPa (**b**).

Under more severe (hydrothermal) conditions of FAL hydrogenation (150 °C, 3.0 MPa), water was involved in the hydrolytic reactions of furan ring opening and Piancatelli rearrangement yielding cyclopentanone (CPONE), which was further hydrogenated to cyclopentanol (CPOL) (see Scheme 1). It should be noted that the route of ring opening to 4-oxopentanal and 5-hydroxy-2-pentanone, which is realized, for example, in the palladium-catalyzed aqueous-phase hydrogenation of FAL under the same conditions [40,41], turned out to be almost completely suppressed when Ni@NiAlO$_x$ catalysts were used. As can be seen from Table 5, the investigated Ni@NiAlO$_x$ catalysts showed a fairly high activity in the hydrogenation, significantly exceeding the supported 10%Ni/γ-Al$_2$O$_3$ reference catalyst in terms of FAL conversion. It should be noted that a higher conversion for a sample of the same composition was achieved [42] using alumina with a specific surface area of more than 400 m$^2$·g$^{-1}$ and carrying out the reaction at a higher hydrogen pressure of 4 MPa. Catalysts with a Ni/Al atomic ratio of 4, as well as those reduced at 600 °C, contained an increased fraction of metallic nickel and made it possible to achieve complete conversion of FAL. Moreover, the catalyst composition (Ni/Al atomic ratio), reduction temperature, and, consequently, the fraction of metallic nickel in Ni@NiAlO$_x$ system strongly affected the selectivity to the products of FAL conversions (Figure 9b). The higher the fraction of metallic nickel (according to XRD data), the lower the selectivity to FOL and CPONE, but the higher the selectivity to CPOL. When the fraction of metallic nickel in the catalysts was more than 40%, FAL and FOL were completely converted under the chosen reaction conditions, and the main hydrogenation products were CPOL and THFOL.

**Table 5.** Catalytic measurements in hydrogenation of FAL under harsh reaction conditions for Ni@NiAlO$_x$ samples differing in the Ni/Al atomic ratio and reduction temperature.

| Catalyst [1] | Conversion of FAL, % [2] | Selectivity, % [3] | | | |
|---|---|---|---|---|---|
| | | **FOL** | **THFOL** | **CPONE** | **CPOL** |
| Ni@NiAlO$_x$-2-500 | 67 | 38 | 5 | 41 | 0 |
| Ni@NiAlO$_x$-4-500 | >99 | 0 | 23 | 19 | 41 |
| Ni@NiAlO$_x$-2-600 [4] | >99 | 2 | 33 | 43 | 8 |
| Ni@NiAlO$_x$-4-600 [4] | >99 | 0 | 20 | 8 | 45 |
| 10%Ni/$\gamma$-Al$_2$O$_3$ | 37 | 4 | 7 | 59 | 2 |

[1] Reaction conditions: 1.0 g catalyst, 5.0 cm$^3$ FAL, 100 cm$^3$ water, 150 °C, 3.0 MPa, 4 h. [2] According to GC. [3] According to GC. The reaction products also included small amounts of intermediates: 4-hydroxy-2-cyclopentenone, 3-hydroxycyclopentanone, 2-cyclopentenone (identified by NMR spectroscopy). [4] Reaction time 200 min.

The hydrogenation rate, which was determined from the amount of hydrogen consumed per unit time, also depended on the composition of the catalyst and its reduction temperature, and the highest reaction rate was in the presence of the Ni@NiAlO$_x$-4-600 catalyst containing the largest fraction of metallic nickel (49%). It is interesting to note that for the most active catalyst samples, the plots of the dependence of the hydrogen consumption rate on the amount of consumed hydrogen (Figure 10) showed a minimum point. Apparently, when this point was reached, the conversion of FAL was completed, after which the active sites became available for adsorption and conversion of FOL and CPONE remaining in the reaction solution (Figure S9); during hydrogenation of these compounds, hydrogen was consumed with a high rate. This observation requires further kinetic investigation.

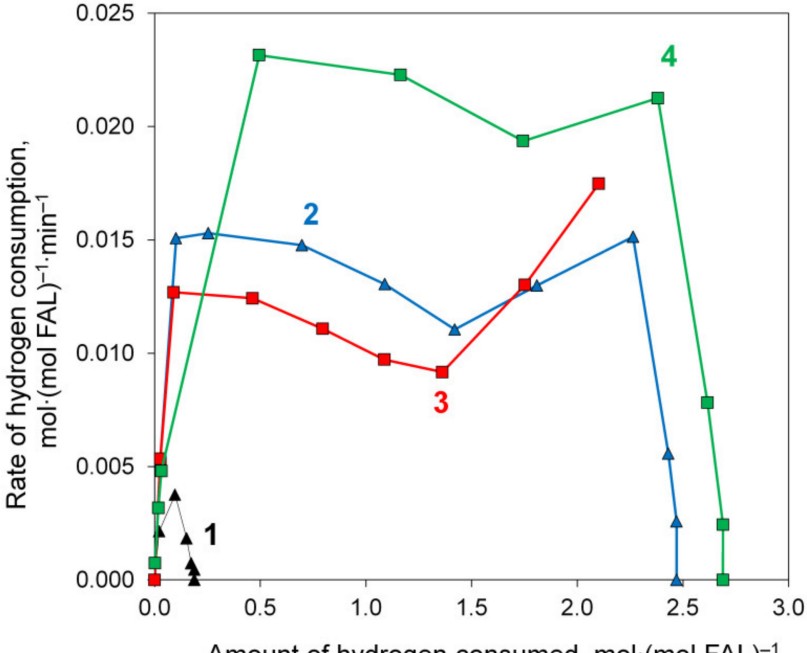

**Figure 10.** Rate of hydrogen consumption plotted vs. amount of hydrogen consumed during aqueous-phase hydrogenation of FAL at 150 °C and 3.0 MPa in the presence of Ni@NiAlO$_x$-2-500 (line 1), Ni@NiAlO$_x$-4-500 (line 2), Ni@NiAlO$_x$-2-600 (line 3), and Ni@NiAlO$_x$-4-600 (line 4).

## 3. Materials and Methods

### 3.1. Catalyst Preparation

The synthesis of carbonate forms of nickel–aluminum LDHs (NiAl-CO$_3$) was based on co-precipitation of double hydroxides from solutions of nitrates upon their interaction with

the solutions containing carbonate and hydroxide ions. LDHs with the Ni:Al atomic ratio in the range of 2–4 were obtained by varying the concentrations of metals in the solutions. The synthesis was carried out at pH 9, which is optimal for precipitation of double hydroxides with the specified composition, and a temperature of 60 °C [8,18,19]. The precipitate was washed with a large amount of distilled water, filtered, and dried for 16 h at 120 °C. To obtain the $NiAlO_x$-$n$ ($n$ is Ni:Al atomic ratio in the range of 2–4) oxide phase, $NiAl$-$CO_3$ samples were calcined at 600 °C. The elemental analysis of LDHs was performed by means of ICP-AES on a Varian 710-ES instrument (Agilent Technologies, Santa Clara, CA, USA), and AAS on a AA-6300 instrument (Shimadzu, Kyoto, Japan).

The reductive treatment of the samples was carried out in a glass reactor in flowing hydrogen (with a rate of 15 mL·min$^{-1}$) at a uniform temperature rise of 6 deg.·min$^{-1}$. Exposure at a given temperature (500 or 600 °C) was 2 h. The reduced samples, cooled in an argon flow, were transferred to the reactor. Designations of the resulting samples of metallic Ni phase dispersed in the oxide matrix $Ni@NiAlO_x$-$n$-$T$ indicate the Ni:Al atomic ratio ($n$) and the reduction temperature ($T$).

Nickel reference catalysts were synthesized by the authors using the conventional method impregnation of supports with a nickel nitrate solution. The supports were represented by γ-alumina (Condea Chemie GmbH, $S_{BET}$ = 202 m$^2$·g$^{-1}$, $D_{av}$ = 10.8 nm) and magnesium–aluminum oxide ($S_{BET}$ = 242 m$^2$·g$^{-1}$, $D_{av}$ = 10.4 nm), which was obtained by calcination of the carbonate form of MgAl LDH (Mg:Al = 3). The samples were calcined at 600 °C and reduced in flowing hydrogen at 600 °C. The content of nickel was comparable with that in the catalysts on the basis of nickel–alumina LDH systems (10%, 50%Ni/γ-Al$_2$O$_3$ and 10%Ni/MgAlO$_x$).

### 3.2. Characterization

XRD studies were performed on a D8 Advance (Bruker, Karlsruhe, Germany) diffractometer using monochromatized CuK$_\alpha$ radiation. Diffraction patterns for the initial samples were recorded over the range of $2\theta$ diffraction angles from 5 to 80° with a scanning step 0.05° and a signal integration time 5 s/step, while for oxide phases, in the $2\theta$ range from 5 to 100° with a scanning step 0.05° and a signal integration time 10 s/step. Unit cell parameters of the initial samples were calculated by the formulas $c = 3d_{003}$ and $a = 2d_{110}$ [15,16]. For quantitative analysis of the identified phases, experimental diffraction patterns were simulated using Rietveld refinement in Topas 4.2 (Bruker) software [43].

Thermal decomposition of the LDH samples was studied by means of thermal analysis TG-DTG-DTA. The measurements were made on an STA-449C Jupiter (Netzsch-Gerätebau GmbH, Selb, Germany) instrument in dynamic mode in an air medium at a heating rate of 10 °C·min$^{-1}$.

Measurements of the nitrogen adsorption–desorption isotherms at 77.4 K were performed using a volumetric vacuum static setup ASAP-2020M (Micromeritics, Norcross, GA, USA). The range of equilibrium-relative pressures was from $10^{-3}$ to 0.996 $P/P_0$. Specific surface area was calculated according to the BET method ($S_{BET}$) in the range of equilibrium-relative pressures of nitrogen vapor $P/P_0$ = 0.05–0.25 from the adsorption isotherm. Differential characteristics of the pore size distribution curves were obtained by the BJH method for the adsorption branch of isotherms. The calculation was based on a cylindrical model of unconnected pores [44].

The dynamics of nickel reduction from its oxide species was investigated using the TPR method on an AutoChem-2920 (Micromeritics, Norcross, GA, USA) chemisorption analyzer equipped with a thermal conductivity detector (TCD). The samples obtained by calcination of NiAl LDH in air at 600 °C were employed for TPR. TPR was carried out with a 10 °C·min$^{-1}$ heating rate using a 10 vol % H$_2$–Ar gas mixture (the flow rate of 30 mL·min$^{-1}$). The reduction of the corresponding oxide phases was conducted up to 900 °C for obtaining the overall TPR profile, or to the fixed temperatures of 550 and 650 °C for measuring the amount of consumed hydrogen and estimating the reduction degree of nickel under the given conditions.

Surface morphology of NiAl LDH and the corresponding oxide was investigated by scanning electron microscopy (SEM) on a JSM-6460 LV (JEOL, Tokyo, Japan) instrument with the tungsten cathode and an accelerating voltage of 20 kV. TEM images were recorded on a JEM-2100 (JEOL, Tokyo, Japan) electron microscope with the lattice resolution 0.14 nm at accelerating voltage 200 kV. The EDX analysis was performed using an INCA 250 spectrometer (Oxford Instruments High Wycombe, UK).

The Ni K-edge XAFS spectra were obtained using a standard transmission mode at beamline 8 (Synchrotron and Terahertz Radiation Center, Institute of Nuclear Physics SB RAS, Novosibirsk, Russia) within the energy range of 8200–9100 eV, which corresponds to the wave vector range up to 15 Å$^{-1}$. The energy of Ni K-edge X-ray absorption was 8333 eV. A channel-cut Si(111) monochromator was used in the study. The EXCURVE software (version 98, Daresbury, Warrington, Cheshire, UK, 1998) package was used to analyze the data.

### 3.3. Catalytic Experiments

Liquid-phase hydrogenation of FAL (99%, Sigma-Aldrich, St. Louis, MO, USA) by molecular hydrogen (99.99%, NCCP, Novosibirsk, Russia) in the presence of the synthesized catalysts was studied in a steel autoclave equipped with an external jacket. Before experiments, FAL was purified by distillation. All catalysts were reduced prior to experiments and transfer to autoclave in an argon atmosphere. In general, freshly reduced catalyst sample (0.5 or 1.0 g; the grain size of 0.25–0.50 mm) was placed in an autoclave together with a certain amount of distilled water. Then, a solution of FAL (5.0 cm$^3$) in water was loaded in the autoclave. The reactor system was flushed with argon and hydrogen to remove residual air. The reaction mixture was heated to a specified temperature with circulation of the liquid heat carrier through the jacket. The hydrogenation was carried out at a temperature of 90 or 150 °C and a total pressure of 2.0 or 3.0 MPa, under vigorous stirring of the reaction mixture. The reaction was controlled by measuring the volume of consumed hydrogen with the use of a gas flow rate measurement system. The reaction rate was calculated on the initial linear segment of kinetic curve as the amount of hydrogen consumed per minute during hydrogenation of FAL. Upon termination of the reaction and cooling, the reaction solution was separated from the catalyst by filtering. The reaction products were identified by $^1$H and $^{13}$C NMR spectroscopy using an Avance-400 (Bruker, Fällanden, Switzerland) NMR spectrometer. The quantitative composition of the reaction solution was found by gas chromatography (GC) on a GKh-1000 (Khromos, Moscow, Russia) instrument equipped with a ValcoBond VB-Wax capillary column (60 m × 0.32 mm, polyethylene glycol, stationary phase thickness 0.50 μm) and a flame ionization detector.

The FAL conversion and selectivity to each product were calculated as follows:

$$\text{FAL conversion (\%)} = \frac{\text{moles of reacted FAL}}{\text{moles of initial FAL}} \times 100 \tag{3}$$

$$\text{Product selectivity (\%)} = \frac{\text{moles of product obtained}}{\text{moles of reacted FAL}} \times 100 \tag{4}$$

### 4. Conclusions

NiAl LDHs with different ratio of metals were synthesized, and their transformations in all steps of the nickel catalyst synthesis were studied. The ratio between oxidized and reduced states of nickel in dependence on the nickel content and catalyst reduction temperature was estimated quantitatively using independent methods. In distinction to the catalysts synthesized by conventional impregnation of the preliminarily prepared support with a solution of the active component, the use of NiAl LDH as the catalyst precursor allows for obtaining, even at a high content of active metal, the dispersed particles of supported nickel that are active under mild conditions of the aqueous-phase hydrogenation. In the synthesis of Ni@NiAlO$_x$ catalysts, the oxide support retains a high specific surface area and is characterized by a uniform pore size distribution, the absence of micropores, and

the prevalence of large mesopores with the average diameter of 13 nm, which is important for providing the accessibility of active component to reagents.

If a lower reduction temperature of 500 °C is used, the size of the reduced particles (estimated from XRD data) is 4–6 nm, irrespective of the nickel content. Therewith, an increase in the amount of such particles with the growth of nickel content in the sample results in a monotonic increase in the furfural conversion. Elevation of the reduction temperature up to 600 °C increased not only the fraction of nickel in metallic state but also its particle size to 10 nm.

The highest activity in hydrogenation of furfural was observed for the catalysts with the Ni/Al atomic ratio of 4. Thus, under mild reaction conditions (90 °C, 2.0 MPa), the conversion reached 93 and 98%, and the hydrogen consumption rates were 78 and 66 (mmol $H_2$)·$g_{cat}^{-1}$·min$^{-1}$ at the catalyst reduction temperatures of 500 and 600 °C, respectively. A lower reduction temperature and the presence of smaller nickel particles led to a higher hydrogenation rate and the formation of furfuryl alcohol as the main product with the selectivity of 70%. High activity of the synthesized nickel catalysts under mild reaction conditions made it possible to perform hydrogenation of furfural using aqueous solvent without a noticeable involvement of water in the transformation of furfural with the formation of additional products.

Under harsh reaction conditions (150 °C, 3.0 MPa), hydrolytic furan ring opening and rearrangement yielding cyclopentanone and cyclopentanol occurred in the presence of Ni@NiAlO$_x$ catalysts. If the catalysts reduced at 600 °C with Ni/Al atomic ratio of 4 were used, then complete conversion of furfural was achieved. In the presence of these active catalysts (containing the largest fraction of metallic nickel), furfural and furfuryl alcohol were completely converted, and cyclopentanol and tetrahydrofurfuryl alcohol were the main hydrogenation products.

Since the formation and size variation of dispersed nickel particles occur under high-temperature activation conditions, this allows for the application of the regularities established in this work, not only for the liquid-phase reactions but also for the high-temperature hydrogenolysis of C–H and C–C bonds. In the future, we also plan to optimize the composition of a Ni@NiAlO$_x$ catalyst and conditions for its obtaining, as well as the conditions of catalytic reaction, which will make it possible to produce one or another target product from FAL with a high yield. The subsequent study of the stability of the optimal catalyst is necessary to investigate the possibility of its further practical use in hydrogenation of FAL.

**Supplementary Materials:** The following supporting information can be downloaded at https://www.mdpi.com/article/10.3390/catal12060598/s1, Figure S1: DTG profiles of NiAl-LDHs with the Ni/Al atomic ratios of 2, 3, and 4. Figure S2: The adsorption–desorption isotherms for the samples with Ni/Al = 2 and 4. Figure S3: Pore size distribution curves calculated by the BJH method from the adsorption branch for NiAlO$_x$ samples obtained from NiAl LDHs with different Ni/Al atomic ratios. Figure S4: TEM image of a Ni@NiAlO$_x$-4-600 sample reduced at 600°C (a); the area of the Ni nanoparticle is marked with a square (b); FFT obtained from the TEM image of the crystal lattice (c). Figure S5: TEM images of the Ni@NiAlO$_x$-4-600 sample at various magnifications. Figure S6: Hydrogen consumption curves vs. time and hydrogen consumption rate vs. volume of hydrogen consumed recorded during aqueous-phase hydrogenation of FAL (90 °C, 2.0 MPa) over 50%Ni/γ-Al$_2$O$_3$ preliminarily calcined at 600 °C and reduced at 600 °C. Figure S7: TEM image of 10%Ni/MgAlO$_x$ sample (Mg:Al = 3, calcination temperature of 600 °C, reduction temperature of 600 °C). Figure S8: The volume of consumed hydrogen vs. time in hydrogenation of FAL on the Ni@NiAlO$_x$ catalysts. Figure S9: Reaction composition time profile of FAL hydrogenation over the Ni@NiAlO$_x$-4-500 catalyst at a temperature of 150 °C and pressure of 3.0 MPa. The concentration of components in the reaction solutions was found by GC. Table S1: Main textural characteristics of samples differing in the Ni/Al atomic ratio and thermal treatment conditions. Table S2. TPR data for NiAlO$_x$ samples (pre-calcination temperature 550 °C). Table S3: Results of EDX analysis of various regions of the Ni@NiAlO$_x$-4-600 sample on TEM images with the calculation of the atomic ratio of elements.

**Author Contributions:** Conceptualization, O.B.B. and V.A.L.; methodology, O.B.B. and V.A.L.; investigation, R.M.M., T.I.G., M.V.T., I.V.M., S.V.T. and V.V.Z.; writing—original draft preparation, O.B.B., R.M.M. and S.V.T.; writing—review and editing, O.B.B., R.M.M. and V.A.L.; supervision, O.B.B. and V.A.L. All authors have read and agreed to the published version of the manuscript.

**Funding:** The work was supported by the Ministry of Science and Higher Education of the Russian Federation within the governmental order for Boreskov Institute of Catalysis (project AAAA-A21-121011490008-3) and Nikolaev Institute of Inorganic Chemistry SB RAS (project 121031700313-8).

**Data Availability Statement:** Data are contained in the article and Supplementary Materials. Any additional data are available on request from the corresponding author.

**Acknowledgments:** The authors thank O.V. Maevskaya, S.V. Vysotsky, S.N. Evdokimov, A.V. Babenko, and R.R. Izmailov for the synthesis of the samples and the study of their composition and catalytic properties; V.V. Kriventsov for his help in XAS measurements; and L.N. Stepanova for the figures design. The studies were carried out using facilities of the shared research center National Center of Investigation of Catalysts at Boreskov Institute of Catalysis, Omsk Regional Center of Collective Usage SB RAS, and the shared research center SSTRC on the basis of the Novosibirsk VEPP-4-VEPP-2000 complex at BINP SB RAS.

**Conflicts of Interest:** The authors declare no conflict of interest.

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
