# Peer review of "Catalysts Derived from Nickel-Containing Layered Double Hydroxides for Aqueous-Phase Furfural Hydrogenation"

_catalysts, doi:10.3390/catal12060598_

Round 1
Reviewer 1 Report
Dear Editor,
Dear Authors,
I read the manuscript careful and unfortunately I cannot recommend its publication in this form. My major concerns are related to catalyst preparation and characterization. The catalytic activity is size and shape, crystallinity , distribution of Ni on the surface or into the wall of the LDHs, etc.
Also, maybe it would be great to have a reference (catalyst without LDHs) or catalyst deposited on a inert support and also pure LDHs. There are many LDHs with catalytic activity!
Author Response
Response to Reviewer 1 Comments
I read the manuscript careful and unfortunately I cannot recommend its publication in this form. My major concerns are related to catalyst preparation and characterization. The catalytic activity is size and shape, crystallinity , distribution of Ni on the surface or into the wall of the LDHs, etc.
Also, maybe it would be great to have a reference (catalyst without LDHs) or catalyst deposited on a inert support and also pure LDHs. There are many LDHs with catalytic activity!
Our general response:
The authors thank the Reviewer for a careful analysis of the presented material. And we regret that we could not clearly show the novelty and originality of our work.
The reviewer is absolutely right that many LDHs have catalytic activity. That is why the number of studies related to these objects is increasing every year. Nickel-containing systems are also being studied extensively, typically in high-temperature reactions such as CO2 methanation (or dry reforming of biogas), methanol steam reforming, growth of CNTs etc.
A feature of the proposed catalyst is: i) the possibility of its operation in mild conditions at temperatures below 100°C (which is not typical for nickel catalysts); ii) the possibility of its operation in water (the most environmentally friendly solvent); iii) atypical reaction route producing significant amounts of cyclopentanol using slightly more severe hydrothermal conditions.
It is clear that the proposed catalytic system is promising, but far from real use. We believe that the main value of our work at this stage is the consistent consideration of all stages of the synthesis of catalysts. We also considered the effect of the Ni/Al ratio in LDH and reduction temperature on the preparation of an active catalyst. We observed the formation of the active metal and the change in the structure of the support during the synthesis using modern physico-chemical methods. Particular attention is paid to determining the quantitative ratio of the oxidized and metallic states of nickel in the finished catalyst depending on the precursor composition and thermal activation conditions.
This type of catalyst, when the active component is generated from a mixed oxide phase, is very different from conventional supported LDH based catalysts. We have prepared and studied such catalysts using LDH as a support precursor and platinum, palladium, and gold as an active component [1-5]. And the regularities between the catalytic activity and the size, shape, electronic state of the deposited metal were established. The described nickel catalysts, in which the active metal is initially introduced into the support matrix and participates in the process of topological transformation in a hydrogen atmosphere, are difficult to study using the traditional algorithm. There are problems in determining the size and shape of nickel particles in such catalysts by traditional methods of electron microscopy or chemisorption methods; it is difficult to quantitatively differentiate nickel particles in the oxidized and reduced state on the surface. As a result, it is difficult to correctly assess their effect on activity, as well as to calculate the specific activity of nickel.
Nevertheless, we believe that systems of this type can be considered as an alternative to expensive palladium catalysts, especially in a renewable feedstock processing scheme. In addition, the established regularities can be applied to the preparation and use of nickel catalysts for other reactions, including high-temperature reactions. Therefore, this article may be of interest to researchers in the field of catalysis.
[1] Belskaya O.B., Zaikovskii V.I., Gulyaeva T.I., Talsi V.P., Trubina S.V., Kvashnina K.O., Nizovskii A.I., Kalinkin A.V., Bukhtiyarov V.I., Likholobov V.A. The effect of Pd(II) chloride complexes anchoring on the formation and properties of Pd/MgAlOx catalysts. Journal of Catalysis. 2020. V.392. P.108-118. DOI: 10.1016/j.jcat.2020.09.021
[2] Belskaya O.B., Likholobov V.A. Development of Approaches to the Formation of Platinum Sites with Desired Properties Using Layer-Structured Supports. Russian Journal of General Chemistry. 2020. V.90. N3. P.495-508. DOI: 10.1134/s1070363220030263
[3] Belskaya O.B., Stepanova L.N., Nizovskii A.I., Kalinkin A.V., Erenburg S.B., Trubina S.V., Kvashnina K.O., Leont’eva N.N., Gulyaeva T.I., Trenikhin M.V., Bukhtiyarov V.I., Likholobov V.A. The Effect of Tin on the Formation and Properties of Pt/MgAl(Sn)Ox Catalysts for Dehydrogenation of Alkanes. Catalysis Today. 2019. V.329. NSI. P.187-196. DOI: 10.1016/j.cattod.2018.11.081
[4] Belskaya O.B., Stepanova L.N., Gulyaeva T.I., Erenburg S.B., Trubina S.V., Kvashnina K., Nizovskii A.I., Kalinkin A.V., Zaikovskii V.I., Bukhtiyarov V.I., Likholobov V.A. Zinc Influence on the Formation and Properties of Pt/Mg(Zn)AlOx Catalysts Synthesized from Layered Hydroxides. Journal of Catalysis. 2016. V.341. P.13-23. DOI: 10.1016/j.jcat.2016.06.006
[5] Stepanova L.N., Belskaya O.B., Trenikhin M.V., Leont’eva N.N., Gulyaeva T.I., Likholobov V.A. Effect of Pt(Au)/MgAlOx catalysts composition on their properties in the propane dehydrogenation. Catalysis Today. 2021.N378. P.96-105. DOI: 10.1016/j.cattod.2021.04.003

Reviewer 2 Report
This manuscript provides a detailed analysis of NiAl layered double hydroxides catalysts for conversion of aqueous-phase furfural hydrogenation. This paper analyzes the NiAl structure and Ni particle size in details. It is a good paper worthy of being published with abundant experimental data. However, there are some issues should be addressed before its accepted.
- The authors mention that this reaction requires a reducing catalyst. Please supplement the reduction conditions for reducing the catalyst.
- Please change the sample’s name in some places. The sample’s name may not be replaced by number, which would be easily confused with the atomic ratio. For example the description of " (b) with the Ni:Al atomic ratio of 2 (1), 3 (2), and 4 (3)" in Figure 1 cannot tell whether it is a legend or a ratio.
- In TPR analysis, the authors only simply quoted the literature, and did not analyze the graph in depth. They mentioned that the hydrogen consumption of line 1 is close to complete reduction. Could the authors provide all of the hydrogen consumption and the degree of reduction?
- In Figure 8, the HRTEM image is unclear. How did the authors find out that is Ni or NiO? Could you please give a partial enlargement or other clear pictures?
- The authors have compared the catalytic activity on their own catalysts. However, please list the literature for comparison to set off the excellent catalytic activity of NiAl LDHs.
- Why choose a different catalyst at 90 °C, 2.0 MPa and 150 °C, 3.0 MPa? Could you please give all the data for a better comparison or explain why choose different catalyst?
- The authors say that the conversion is complete at 150 °C and 3.0 MPa. Is the catalyst still in the LDHS structure after the reaction under 150 °C and 3.0 MPa, and would the Ni nanoparticles agglomerate or not? In the introduction, the authors mentioned that “the NiAl mixed oxides 64 obtained by calcination of carbonate-containing NiAl LDHs (takovite) demonstrate good catalytic properties owing to high dispersion of the particles and thermal stability”. However, the stability or cycle test data is not shown in the manuscript. Please provide corresponding data to demonstrate their stability for aqueous-phase furfural hydrogenation.
- In Figure S3, what is unit of HM? In Figure S8, how does the authors calculate the concentration of every product? In addition, please provide the calculation formula for the conversion, product selectivity and reaction rate.
- In the introduction, the authors introduced the hydrogenation reactions. Some recent related references are suggested to be cited ( https://doi.org/10.3390/catal12030320; https://www.science.org/doi/10.1126/science.abm9257; https://doi.org/10.1016/j.fuel.2020.119151; https://doi.org/10.1007/s40789-021-00444-2 ).
Author Response
Response to Reviewer 2 Comments
This manuscript provides a detailed analysis of NiAl layered double hydroxides catalysts for conversion of aqueous-phase furfural hydrogenation. This paper analyzes the NiAl structure and Ni particle size in details. It is a good paper worthy of being published with abundant experimental data. However, there are some issues should be addressed before its accepted.
Our general response:
We are grateful to the reviewer for his/her positive overall evaluation of our article. We tried to take into account all the comments and, in accordance with them, made changes into the manuscript. In some cases, we also provided explanations that clarify our point of view
- The authors mention that this reaction requires a reducing catalyst. Please supplement the reduction conditions for reducing the catalyst.
Response 1:
Thank you for your recommendation. We have added the required information in Materials and Methods, page 3.
The reductive treatment of the samples was carried out in a glass reactor in flowing hydrogen (with a rate of 15 ml×min-1) at a uniform temperature rise of 6 deg.×min-1. Exposure at a given temperature (500 or 600°C) was 2 hours. The reduced samples, cooled in an argon flow, were transferred to the reactor.
- Please change the sample’s name in some places. The sample’s name may not be replaced by number, which would be easily confused with the atomic ratio. For example the description of " (b) with the Ni:Al atomic ratio of 2 (1), 3 (2), and 4 (3)" in Figure 1 cannot tell whether it is a legend or a ratio.
Response 2:
This is a really useful note. We have unified the designations for all figures (Figures 1, 3, 4, 7, 10) with indication of curve numbers (line 1, line 2, etc.)
- In TPR analysis, the authors only simply quoted the literature, and did not analyze the graph in depth. They mentioned that the hydrogen consumption of line 1 is close to complete reduction. Could the authors provide all of the hydrogen consumption and the degree of reduction?
Response 3:
Thank you for your advice. We have added quantitative data regarding the reduction of mixed oxides
According to Figure 3, as the fraction of nickel increases, the fraction of hydrogen consumed in the low-temperature region also increases, which produces changes in the shape of the peak and position of the maximum. The hydrogen consumption maximum shifts from 600 °C for NiAlOx-2 and NiAlOx-3 (Ni:Al = 2 and 3) to 500 °C for NiAlOx-4 (Ni:Al = 4). Thus, an increase in the fraction of nickel-enriched oxide phase weakens the Ni‒Al interaction, facilitates the reduction of metal, and makes it possible to obtain metallic nickel under milder conditions. Data on the amount of consumed hydrogen and the degree of nickel reduction depending on the composition of mixed oxide samples and TPR conditions are given in Table S2. For the most “high-temperature” sample with the minimum nickel content (Ni:Al = 2), measurements of the hydrogen amount consumed in TPR experiment showed that upon reduction at 550 °C (holding for 1 h) the reduction degree of nickel did not exceed 55% (according to the equation (2)). When TPR was performed up to 900 °C, as in the case of holding at 650 °C, hydrogen consumption was close to stoichiometric for the complete reduction of metal.
Table S2. TPR data for NiAlOx samples (pre-calcination temperature 550 °C).
Sample |
Tmax of TPR, oC |
Total amount of consumed H2, mmol×g-1* |
Reduction degree, %** |
NiAlОх-2 |
500 |
4.9 |
55 |
NiAlОх-2 |
650 |
9.6 |
105 |
NiAlОх-2 |
900 |
10.9 |
120 |
NiAlОх-3 |
650 |
9.8 |
93 |
NiAlОх-3 |
900 |
11.5 |
108 |
NiAlОх-4 |
650 |
11.5 |
105 |
NiAlОх-4 |
900 |
11.9 |
108 |
* The observed some excess amount of consumed hydrogen is probably due to the solubility of hydrogen in metallic nickel (R.B. McLellan, P.L. Sutter, Thermodynamics of the hydrogen-nickel system, Acta Metall. 32 (1984) 2233e2239).
** evaluated from reaction: NiO+H2®Ni+H2O
- In Figure 8, the HRTEM image is unclear. How did the authors find out that is Ni or NiO? Could you please give a partial enlargement or other clear pictures?
Response 4:
We are grateful to the reviewer for this comment.
Changes have been made to Figure 8. We identified the particles observed in TEM images by their crystal lattices: an interplanar spacing of 0.203 nm is characteristic of the (011) crystalline planes of metallic nickel (PDF 45-1027), an interplanar spacing of 0.208 nm is characteristic of the (012) crystalline planes of the NiO phase (PDF 44- 1159). Calculations of interplanar distances in nanoparticles were carried out using the Digital Micrograph “Gatan” software after Fast Fourier Transform (FFT) processing of TEM images (Figure S4).
Figure 8. TEM images of the Ni@NiAlOx-4-600 sample at different magnification (a, b); crystal lattice of NiO (c) and Ni (d) nanoparticles.
Figure S4. TEM image of a Ni/NiAlOx-4-600 sample reduced at 600°C (a); the area of the Ni nanoparticle is marked with a square (b); FFT obtained from the TEM image of the crystal lattice (c).
- The authors have compared the catalytic activity on their own catalysts. However, please list the literature for comparison to set off the excellent catalytic activity of NiAl LDHs.
Response 5:
We thank the reviewer for this comment. One of the latest reviews Y. Wang, D. Zhao, D. Rodríguez-Padrón, C. Len. Recent Advances in Catalytic Hydrogenation of Furfural. Catalysts 2019, 9, 796; doi:10.3390/catal9100796 contains information on the main types of furfural hydrogenation catalysts.
As for nickel-containing catalysts, they are mainly represented by systems of nickel on a carbon supports [1, 2], including modified ones [3]. In the aqueous phase, the reaction is carried out at a temperature of 100-120°C and a pressure of 4 MPa to obtain THFA as the main product. When using the i-PrOH solvent, the reaction temperature is 150-260 °C, the pressure is 3-5 MPa, the main product is MF. Nickel catalysts based on modified alumina are also described (140 оС, H2O, 4 MPa, THFA is main product) [4].
Only one of these catalytic systems was close to those given in our work in terms of composition and method of preparation [5]. However, the studies were carried out in flow mode at a temperature of 180 °C in i-PrOH. And unlike nickel catalysts on carbon and alumina support, the reaction proceeded to the formation of furfuryl alcohol (as in our work at 90 °C in water)
For comparison, we can also cite work with a similar composition of catalysts obtained from NiAl-LDH with Ni/Al=2 [6]. However, the reaction was carried out in the gas phase at 155°C, a pressure of about 1 atm, and H2/furfural molar ratio of 25. The main transformation route was the hydrogenation of the aldehyde with the formation of furfuryl alcohol, followed by the formation of side or sequential products such as THFA, furan, and 2-methylfuran. Furfural conversion was 16% (this value is much lower than in our work, when the reaction was carried out at 90°C in water).
In work [7], nickel catalysts supported on mixed NiAlOx oxides (MMO) were prepared, as in our work, by structural-topological transformation (treatment by calcination in an H2 flow) from hydrotalcite precursors (NiAl-LDH) with carbonate or nitrate in the interlayer region (denoted as Ni/MMO-CO3 and Ni/MMO-NO3). Interestingly, the selectivity of furfural hydrogenation can be switched depending on the composition of the interlayer space of LDH: Ni/MMO-NO3 exhibited high selectivity (97%) to furfural alcohol, while Ni/MMO-CO3 exhibited exceptional selectivity (99%) with respect to tetrahydrofurfuryl alcohol. The reaction was carried out in i-PrOH at temperature 110 °C and H2 pressure 3 MPa.
From this brief review, it can be seen that we are following research in this direction, however, an analysis of a number of publications does not yet allow a correct comparison due to a significant difference in the composition of catalysts and their testing conditions. The fact that the nickel catalysts presented in our work make it possible to achieve complete conversion of furfural under mild conditions of aqueous-phase catalysis allows us to conclude that their use is promising.
[1] Gong, W.; Chen, C.; Zhang, H.; Wang, G.; Zhao, H. Highly dispersed Co and Ni nanoparticles encapsulated in N-doped carbon nanotubes as efficient catalysts for the reduction of unsaturated oxygen compounds in aqueous phase. Catal. Sci. Technol. 2018, 8, 5506–5514)
[2] Wang, Y.; Prinsen, P.; Triantafyllidis, K.S.; Karakoulia, S.A.; Trikalitis, P.N.; Yepez, A.; Len, C.; Luque, R. Comparative study of supported monometallic catalysts in the liquid-phase hydrogenation of furfural: Batch versus continuous flow. ACS Sustain. Chem. Eng. 2018, 6, 9831–9844
[3] Gong, W.; Chen, C.; Wang, H.; Fan, R.; Zhang, H.; Wang, G.; Zhao, H. Sulfonate group modified Ni catalyst for highly efficient liquid-phase selective hydrogenation of bio-derived furfural. Chin. Chem. Lett. 2018, 29, 1617–1620
[4] Yang, Y.; Ma, J.; Jia, X.; Du, Z.; Duan, Y.; Xu, J. Aqueous phase hydrogenation of furfural to tetrahydrofurfuryl alcohol on alkaline earth metal modified Ni/Al2O3. RSC Adv. 2016, 6, 51221–51228
[5] Manikandan, M.; Venugopal, A.K.; Prabu, K.; Jha, R.K.; Thirumalaiswamy, R. Role of surface synergistic effect on the performance of Ni-based hydrotalcite catalyst for highly effcient hydrogenation of furfural. J. Mol. Catal. Chem. 2016, 417, 153–162
[6] T. P. Sulmonetti, S. H. Pang, M. T. Claure, S. Lee, D. A. Cullen, P. K. Agrawal, C. W. Jones. Vapor phase hydrogenation of furfural over nickel mixed metal oxide catalysts derived from layered double hydroxides Applied Catalysis A: General 517 (2016) 187–195
[7] X. Meng, Y. Yang, L. Chen, M. Xu, X. Zhang, M. Wei. A Control over Hydrogenation Selectivity of Furfural via Tuning Exposed Facet of Ni Catalysts ACS Catal. 2019, 9, 4226−4235
- Why choose a different catalyst at 90 °C, 2.0 MPa and 150 °C, 3.0 MPa? Could you please give all the data for a better comparison or explain why choose different catalyst?
Response 6:
Preliminary experiments under milder conditions have shown that samples with Ni:Al = 2 and 3 show very similar catalytic properties. Therefore, in further tests under more severe conditions, we excluded samples with an intermediate Ni:Al ratio from consideration. The use of samples with Ni:Al ratios of 2 and 4 made it possible to reveal the influence of both the nickel content and the reduction temperature on the catalytic properties.
- The authors say that the conversion is complete at 150 °C and 3.0 MPa. Is the catalyst still in the LDHS structure after the reaction under 150 °C and 3.0 MPa, and would the Ni nanoparticles agglomerate or not? In the introduction, the authors mentioned that “the NiAl mixed oxides 64 obtained by calcination of carbonate-containing NiAl LDHs (takovite) demonstrate good catalytic properties owing to high dispersion of the particles and thermal stability”. However, the stability or cycle test data is not shown in the manuscript. Please provide corresponding data to demonstrate their stability for aqueous-phase furfural hydrogenation.
Response 7:
We appreciate the reviewer for this comment. As follows from the experimental part, before the catalytic experiments, the precipitated and dried NiAl-LDHs were first calcined at 600 oC and then reduced at 500 or 600 oC. Thus, the catalysts were mixed NiAl oxides (and were not LDHs) containing nickel metal particles (hence, the designation of the catalysts is Ni@NiAlOx). At the same time, it is known that nickel-aluminum oxides obtained from the corresponding LDHs do not have a memory effect and do not restore the layered structure upon contact with water.
In the Introduction, we analyzed literature data and really noted that one of the advantages of catalysts based on NiAl mixed oxides is their stability, especially at high temperatures. At the same time, in our work, we did not study whether the agglomeration of nickel particles occurs during the reaction and whether the catalyst is deactivated. We believe that stability studies would be appropriate for the potential practical application, i.e. if one of the catalysts showed excellent catalytic characteristics, in particular, a high yield (more than 95%) of one of the products, which has not yet been achieved for the studied samples. The aim of this work was to study the genesis of NiAl catalysts and to establish the relationship between the preparation conditions of NiAl catalysts (including Ni:Al ratio) and their catalytic properties. In the future, we plan to study the kinetics of furfural hydrogenation in the presence of these catalysts (as indicated at the end of Section 3.3) and further we plan to optimize the composition of a Ni@NiAlOx catalyst and conditions for its obtaining, as well as the conditions of catalytic reaction, which will make it possible to produce one or another target product from FAL with a high yield. In our opinion, only then it would be highly advisable to study also the stability for an optimal catalyst. We added these remarks to the Conclusions.
- In Figure S3, what is unit of HM?
Response 8:
We are grateful to the reviewer for this comment and regret for our negligence. Axes labels in Figure S3 were corrected.
In Figure S8, how does the authors calculate the concentration of every product? In addition, please provide the calculation formula for the conversion, product selectivity and reaction rate.
As indicated in Section 2.3, the quantitative composition of the reaction solutions was found by gas chromatography. According to the reviewer comment, we noted this in the caption of Figure S9.
The reaction was controlled by measuring the volume of consumed hydrogen with the use of a gas flow rate measurement system. And the reaction rate was calculated on the initial linear segment of kinetic curve as the amount of hydrogen consumed per minute during hydrogenation of FAL. We made this clarification in the text of Section 2.3, page 4. The formulas used in the calculations of furfural conversion and product selectivities were also given in the text of Section 2.3.
- In the introduction, the authors introduced the hydrogenation reactions. Some recent related references are suggested to be cited https://doi.org/10.3390/catal12030320; https://www.science.org/doi/10.1126/science.abm9257; https://doi.org/10.1016/j.fuel.2020.119151; https://doi.org/10.1007/s40789-021-00444-2 ).
Response 9:
We are grateful for providing information about new interesting studies and we have used the most appropriate of them to describe approaches that make it possible to obtain dispersed particles of supported nickel (page 2).
In some approaches, dispersed nickel particles can be obtained by the surfactant templating method using cationic, anionic and non-ionic surfactants as the structure regulating agents [5], also, the choice of Ni-MOF as precursors can provide a high surface area and the formation of isolated small-sized Ni nanoparticles [14].
[14] Ye, R-P.; Liao, L.; Reina, T. R.; Liu J.; Chevella D.; Jin, Y.; Fan, M.; Liu, J. Fuel 2021, 285, 119151. https://doi.org/10.1016/j.fuel.2020.119151

Reviewer 3 Report
The authors investigated the aqueous-phase hydrogenation of furfural over NiAl-LDH derived Ni@NiAlOx catalysts. Herein, product selectivity depends on reaction conditions and fraction of metallic Ni content in Ni@NiAlOx. This is a very interesting topic and the authors have some interesting experimental data. However, there are insufficient data on the relationship between the catalytic process and the structural properties of catalysts. I can recommend it for publication. Followings are my comments.
- For the Ni@NiAlOx-Y-Z00 (Y=2,3,4 and Z= 500, 600oC) catalysts, the catalyst acidic sites should be confirmed by NH3-TPD and Pyridine FT-IR tests.
- Prove relationship between the catalytic performance and the structural properties of catalysts.
- Kindly represent reaction conditions in Scheme 1.
- Highlight the aqueous-phase hydrogenation could give high yield and increase the selectivity under mild reaction conditions compare to vapour-phase hydrogenation (New J. Chem., 2022, 46, 5907) in the Introduction part.
- S5 shows 50%Ni/Al2O3 catalyst with a very low activity. However, Ni/Al2O3 catalyst shows around 99% conversion of furfural with good selectivity (RSC Adv., 2016, 6, 51221). Please add an explanation.
Author Response
Response to Reviewer 3 Comments
The authors investigated the aqueous-phase hydrogenation of furfural over NiAl-LDH derived Ni@NiAlOx catalysts. Herein, product selectivity depends on reaction conditions and fraction of metallic Ni content in Ni@NiAlOx. This is a very interesting topic and the authors have some interesting experimental data. However, there are insufficient data on the relationship between the catalytic process and the structural properties of catalysts. I can recommend it for publication. Followings are my comments.
Our general response:
The authors thank the referee for analyzing the presented material. We fully agree with your specific comments and hope that changing the manuscript, taking into account all the comments, will bring the article closer to the level of the Journal
- For the Ni@NiAlOx-Y-Z00 (Y=2,3,4 and Z= 500, 600oC) catalysts, the catalyst acidic sites should be confirmed by NH3-TPD and Pyridine FT-IR tests.
Response 1:
Thank you for your recommendations. We have examined a number of samples by the temperature-programmed desorption of NH3 method to obtain data on their acidic properties. The obtained TPD profiles and data on the amount of desorbed ammonia are given below. Our data are in good agreement with the published results (for example, Gac, W. Acid–base properties of Ni–MgO–Al2O3 materials. Applied Surface Science 257 (2011) 2875–2880). The density of acid sites is 3.4 and 3.6 µmol m-2 in our work and in the article, respectively, for a sample of similar composition.
Figure. Temperature-programmed desorption of NH3 for g-Al2O3 (line 1), 50%Ni/Al2O3 (line 2), Ni@NiAlOx-2-600 (line 3), Ni@NiAlOx-4-600 (line 4).
Table. Acid properties from NH3 desorption data
Sample |
NH3 desorption, µmol g-1 |
NH3 desorption, µmol m-2 |
g-Al2O3 |
514 |
2.5 |
50%g-Ni/Al2O3 |
385 |
- |
Ni@NiAlOx-2-600 |
421 |
3.4 |
Ni@NiAlOx-4-600 |
418 |
4.4 |
It can be seen that the surface of both aluminum oxide and nickel-aluminum samples has an acidic character. The found direction of change in acidity suggests that the influence of this parameter may be important for controlling the route of furfural conversion and the stability of catalysts.
However, in order to draw correct conclusions, a more thorough and lengthy study is needed. We would like to explore this issue in detail in our future work and not present the acidity analysis in this article (this can significantly increase its length). I hope the reviewer will support us in this decision.
- Prove relationship between the catalytic performance and the structural properties of catalysts.
Response 2:
We thank the reviewer for this comment. The relationships that were obtained in this work and the features of our study are described below.
The main value of our study is the consistent consideration of all stages of catalyst synthesis. We also established the influence of the Ni/Al ratio in LDH and the reduction temperature on the behavior of the catalyst. We observed the formation of an active metal and a change in the structure of the support during synthesis using modern physicochemical methods. Particular attention is paid to determining the quantitative ratio of the oxidized and metallic states of nickel in the finished catalyst depending on the precursor composition and thermal activation conditions. The dependence of the selectivity of the formation of hydrogenation products on the proportion of metallic Ni in the Ni@NiAlOx system for the hydrogenation of FAL was established both under mild and more severe hydrothermal conditions (Figure 9).
It should be noted that the presented type of catalyst, when the active component is created from the mixed oxide phase, is very different from conventional supported catalysts based on LDH. Previously, we prepared and studied catalysts using LDH as a support precursor and Pt, Pd and Au as an active component, and we established the dependences of the catalytic activity on the size, shape, and electronic state of the supported metal.
The nickel catalysts described here, in which the active metal is initially introduced into the support matrix and participates in the process of topological transformation in a hydrogen atmosphere, are difficult to study using the traditional algorithm. There are problems in determining the size and shape of nickel particles in such catalysts by conventional electron microscopy or chemisorption methods; it is difficult to quantitatively distinguish between nickel particles in the oxidized and reduced state on the surface. As a result, it is difficult to correctly assess their effect on activity, as well as to calculate the specific activity of nickel.
In addition, as features of the obtained nickel catalysts, one can note the possibility of their operation under mild conditions at temperatures below 100°C; their ability to work in water; an atypical reaction route to form significant amounts of cyclopentanol using slightly more severe hydrothermal conditions.
- Kindly represent reaction conditions in Scheme 1.
Response 3:
We thank the reviewer for this comment. Scheme 1 is a generalized reaction network that is valid for various conditions. Cyclopentanone and cyclopentanol are formed with a noticeable yield only under harsh hydrothermal conditions (150°C, 3.0 MPa) when water is involved in catalytic conversions. Under these conditions, furfuryl alcohol and tetrahydrofurfuryl alcohol are also formed, but are most often not major products (see Table 5) and dominate under milder reaction conditions (see Table 4). We made a corresponding clarification in the scheme caption
- Highlight the aqueous-phase hydrogenation could give high yield and increase the selectivity under mild reaction conditions compare to vapour-phase hydrogenation (New J. Chem., 2022, 46, 5907) in the Introduction part.
Response 4:
Thank you for your recommendations. We have expanded the introduction by pointing to this study.
The synthesized catalysts were studied in the aqueous-phase hydrogenation of furfural (FAL) (FAL is considered as a representative of platform molecules). The use of mild aqueous-phase reaction conditions is important for processing of vegetable feedstock and initiates the development of advanced, efficient and more available and non-noble metal-based catalysts [33].
[33] Pothu, R.; Gundeboyina, R.; Boddula, R.; Perugopu, V.; Ma, J. Recent advances in biomass-derived platform chemicals to valeric acid synthesis New J. Chem. 2022, 46, 5907-5921. https://DOI: 10.1039/d1nj05777d]
- S5 shows 50%Ni/Al2O3 catalyst with a very low activity. However, Ni/Al2O3 catalyst shows around 99% conversion of furfural with good selectivity (RSC Adv., 2016, 6, 51221). Please add an explanation.
Response 5:
Indeed, according to [44], the conversion of furfural for 10%Ni/Al2O3 was above 99% (with the formation of products of suboptimal composition - almost 40% accounted for 2-methyltetrahydrofuran and unidentified products). The differences between these results and ours are explained both by the difference in the reaction conditions (a higher hydrogen pressure of 4 MPa was used in [44]) and by the features of the catalyst synthesis. In [44], alumina obtained by the authors (without a detailed determination of its composition and structure) with a specific surface area of more than 400 m2×g-1 was used. In our work, for a correct comparison of samples, we used gamma aluminum oxide (Condea Chemie GmbH), which is characterized by a minimum amount of impurities and a surface area of SBET = 202 m2∙g-1, close to the surface of NiAl-mixed oxides. 180 m2 g-1.
However, we have added new information to our article as follows:
It should be noted that a higher conversion for a sample of the same composition was achieved [44] using alumina with a specific surface area of more than 400 m2×g-1 and carrying out the reaction at a higher hydrogen pressure of 4 MPa.
[44] Yang, Y.; Ma, J.; Jia, X.; Du, Z.; Duan, Y.; Xu,J. Aqueous phase hydrogenation of furfural to tetrahydrofurfuryl alcohol on alkaline earth metal modified Ni/Al2O3. RSC Adv. 2016, 6, 51221-51228. https://DOI: 10.1039/c6ra05680f
